# Small Molecule 20S Proteasome Enhancer Regulates MYC Protein Stability and Exhibits Antitumor Activity in Multiple Myeloma

**DOI:** 10.3390/biomedicines10050938

**Published:** 2022-04-19

**Authors:** Evert Njomen, Allison Vanecek, Theresa A. Lansdell, Ya-Ting Yang, Peter Z. Schall, Christi M. Harris, Matthew P. Bernard, Daniel Isaac, Omar Alkharabsheh, Anas Al-Janadi, Matthew B. Giletto, Edmund Ellsworth, Catherine Taylor, Terence Tang, Sarah Lau, Marc Bailie, Jamie J. Bernard, Vilma Yuzbasiyan-Gurkan, Jetze J. Tepe

**Affiliations:** 1Department of Chemistry, Michigan State University, East Lansing, MI 48824, USA; enjomen@scripps.edu (E.N.); vaneceka@chemistry.msu.edu (A.V.); hemmingc@msu.edu (C.M.H.); 2Department of Pharmacology & Toxicology, Michigan State University, East Lansing, MI 48824, USA; lansdel1@msu.edu (T.A.L.); mbernard@msu.edu (M.P.B.); giletto1@msu.edu (M.B.G.); ellswo59@msu.edu (E.E.); marc.bailie@inds-inc.com (M.B.); jbernard@msu.edu (J.J.B.); 3Comparative Medicine and Integrative Biology Program, Michigan State University, East Lansing, MI 48824, USA; yangyat1@msu.edu (Y.-T.Y.); schallpe@msu.edu (P.Z.S.); 4Breslin Cancer Center, Division of Hematology/Oncology, Michigan State University, Lansing, MI 48910, USA; isaacdan@msu.edu (D.I.); oalkharabsheh@health.southalabama.edu (O.A.); anas.al-janadi@spectrumhealth.org (A.A.-J.); 5Department of Biology, University of Waterloo, Waterloo, ON N2L 3G1, Canada; catherine000taylor@gmail.com (C.T.); terence.tang@uwaterloo.ca (T.T.); sarah.lau@uwaterloo.ca (S.L.); 6Department of Microbiology and Molecular Genetics, Michigan State University, East Lansing, MI 48824, USA; 7Department of Small Animal Clinical Sciences, Michigan State University, East Lansing, MI 48824, USA

**Keywords:** multiple myeloma, MYC, proteosome, novel treatment, chemotherapy, protein degradation

## Abstract

Despite the addition of several new agents to the armamentarium for the treatment of multiple myeloma (MM) in the last decade and improvements in outcomes, the refractory and relapsing disease continues to take a great toll, limiting overall survival. Therefore, additional novel approaches are needed to improve outcomes for MM patients. The oncogenic transcription factor MYC drives cell growth, differentiation and tumor development in many cancers. MYC protein levels are tightly regulated by the proteasome and an increase in MYC protein expression is found in more than 70% of all human cancers, including MM. In addition to the ubiquitin-dependent degradation of MYC by the 26S proteasome, MYC levels are also regulated in a ubiquitin-independent manner through the REGγ activation of the 20S proteasome. Here, we demonstrate that a small molecule activator of the 20S proteasome, TCH-165, decreases MYC protein levels, in a manner that parallels REGγ protein-mediated MYC degradation. TCH-165 enhances MYC degradation and reduces cancer cell growth in vitro and in vivo models of multiple myeloma by enhancing apoptotic signaling, as assessed by targeted gene expression analysis of cancer pathways. Furthermore, 20S proteasome enhancement is well tolerated in mice and dogs. These data support the therapeutic potential of small molecule-driven 20S proteasome activation for the treatments of MYC-driven cancers, especially MM.

## 1. Introduction

Multiple myeloma (MM) is a malignant neoplasm, characterized by the abnormal proliferation of plasma cells. As recently reviewed [1], current treatment strategies now include immunomodulatory drugs (lenalidomide, pomalidomide), proteasome inhibitors (bortezomib, carfilzomib, ixazomib), monoclonal antibodies directed at CD38 (elotuzumab, daratumumab and isatuiximab) and most recently, a B-cell maturation antigen directed genetically modified autologous chimeric antigen receptor (CAR)T cell therapy (idecabtagene vicleucel), available for certain patients. Autologous stem cell transplantation is another option. With these approaches, patient survival has increased from a few years in the 1990s to more than eight years. However, treatment regimens are challenging, relapse is experienced by the majority of patients, and novel approaches are needed.

Overexpression of MYC is found in more than 70% of all human cancers, with a high prevalence in cancers of hematopoietic origin, such as acute lymphoblastic leukemia and multiple myeloma [2,3]. In normal cells, MYC protein levels are kept in check through multiple post-translational modifications that alter susceptibility toward proteasomal degradation [4]. Dysregulation of MYC clearance can induce its oncogenic transcriptional activity, resulting in the transcription of multiple tumor-promoting genes [5]. Due to its highly dynamic intrinsically disordered 3-D conformation, MYC has evaded pharmacological manipulation and remains one of the most sought anticancer targets [6]. Strategies targeting MYC transcription through inhibition of bromodomain and extraterminal (BET) bromodomain proteins that regulate MYC expressions, such as through the small molecules JQ1 [7], OTX015 [8] and CPI-0610 [9], have been explored, as well as those targeting MYC translation, such as CMLD010509 [10]. However, these approaches are still under development and not yet approved for clinical use. Here, we present a novel approach to tackling MYC-driven tumors.

One of the major pathways of MYC degradation involves ubiquitin-dependent degradation by the 26S proteasome [4,11]. In this case, the proteolytic 20S core particle is associated with the 19S caps, to form the fully assembled 26S proteasome [12]. The 19S caps facilitate the recognition of ubiquitinylated proteins, their unfolding and transportation into the 20S core particle [13]. However, recent studies have also implicated the ubiquitin-independent degradation of MYC [14]. The ubiquitin-independent degradation pathway involves the 20S proteasome associated with its activator complex, REGγ (11S cap, encoded by PSME3, proteasome activator subunit 3). The REGγ-20S complex primarily targets unfolded or intrinsically disordered proteins, including unbound MYC [15]. Overexpression of the 20S proteasome activating protein REGγ enhanced MYC protein degradation, and subsequently inhibited MYC-mediated gene expression and cell proliferation [14].

Recently, small molecules have emerged as 20S proteasome activators to prevent the accumulation of proteotoxic intrinsically disordered proteins [16,17,18,19,20,21,22,23,24,25,26]. Herein, we evaluate the effects of 20S proteasome activation by a small molecule, TCH-165, to reduce oncogenic signaling by overexpressing MYC.

TCH-165 is part of a privileged class of imidazoline-based scaffolds [27,28,29,30,31,32,33], and is one of the few known molecules shown to enhance 20S-mediated proteolysis nearly 10-fold (i.e., 1000%) by favoring a proteolytically active, open-gate 20S proteasome subcomplex [21]. Our findings show that TCH-165 effectively reduces MYC protein levels in cell culture, inhibits cancer cell proliferation, and inhibits tumor growth in vivo. Importantly, this proteasome activator also effectively kills primary cells from bortezomib unresponsive MM patients. Moreover, the 20S proteasome activator is well tolerated in vivo (in mice and in dogs) at therapeutic concentrations. These studies support further exploration of 20S proteasome enhancement as a new and well-tolerated approach to treat various disorders in which IDPs play a significant pathogenic role, including MYC-driven oncogenesis.

## 2. Materials and Methods

The in vivo studies were conducted in accordance with the current guidelines for animal welfare [34]. The procedures used in this study have been reviewed and approved by the Michigan State University Institutional Animal Care and Use Committee (IACUC) as well as by the University of Waterloo Animal Care Committee (Waterloo, ON, Canada), as established by the Canadian Council on Animal Care and the Province of Ontario Animals for Research Act. All data are presented in compliance with the ARRIVE guidelines [35].

### 2.1. Cell Culture

RPMI-8226, L363 and NCI-H929 cells were obtained from ATCC (Manassas, VA, USA) and were maintained in RPMI medium supplemented with 10% fetal bovine serum, and 100 U/mL penicillin/streptomycin, at 37 °C with 5% CO_2_. DMSO was used as the compound vehicle or vehicle control at a 0.1% final concentration for both cell culture and enzyme assay.

### 2.2. Primary Cell Culture and Cell Viability

Bone marrow aspirates were obtained by Dr. D. Isaac or by Dr. O. Alkharabsheh, with an IRB-approved protocol, and after obtaining informed patient consent (MSU IRB LEGACY17-267). White blood cells (WBCs) were separated from red blood cells by the Ficoll density gradient. Multiple myeloma cells were then isolated from WBCs using human CD138 affinity magnetic beads (Miltenyi Biotec; cat#130-093-062; Gaithersburg, MD, USA) following the manufacturer’s protocol. Cells were cryopreserved in 10% DMSO in FBS. Revived cells were maintained in an RPMI medium with 10% FBS for 10 days prior to treatment. Primary or established cells (10,000/well) were seeded in 96-well plates in a 100 μL medium and treated with TCH-165 or bortezomib for 72 h. Cells were equilibrated to RT and CellTiter-Glo (Promega; Madison, WI, USA) solution (100 μL) was added and incubated with shaking for 10 min at RT. The assay plate was then allowed to equilibrate for 5 min at RT and luminescent readings were taken on a SpectraMax^®^ M5. Data are presented as a percentage of the vehicle control for each experimental condition, after background subtraction. Error bars represent SD for triplicate wells.

### 2.3. Proteasome Activity in Purified Protein Assay

The proteolytic activity of the 20S proteasome can be monitored in vitro by measuring the cleavage of fluorogenic peptide substrates for the different catalytic sites as an increase in 7-amino-4-methylcoumarin (AMC) fluorescence over time, as previously reported [21]. Purified human 20S proteasome was pre-treated with a concentration range of TCH-165 or DMSO (vehicle control) for 15 min at 37 °C, on a black flat/clear bottom 96-well plate containing 1 nM of human constitutive 20S proteasome in 50 mM Tris-HCl pH 7.5. Fluorogenic substrates, Suc-LLVY-AMC (CT-L activity, 10 μM), Z-LLE-AMC (Casp-L activity, 10 μM), Boc-LRR-AMC (Tryp-L activity, 20 μM), were then added and the enzymatic activity measured at 37 °C on a SpectraMax M5e spectrometer by measuring an increase in the fluorescence unit per minute for 1 h at 380–460 nm. The release of AMC was monitored as fluorescence over time for 1 h and the resulting 20S activity changes were determined by comparing to the untreated 20S and calculating the fold-increase in activity over vehicle control for each analog at a given concentration.

### 2.4. MYC Degradation in MM Cells

Western blot: RPMI-8226 cells were grown to approximately 80% confluency in a T-75 flask. Cells were treated with either vehicle (DMSO) or TCH-165 at the indicated concentrations, for 4 h. In experiments involving bortezomib, cells were pre-treated with bortezomib (5 μM) for 1 h before adding TCH-165 or vehicle for a further 4 h. Cells were pelleted and washed with chilled PBS buffer (2X) and resuspended in chilled RIPA buffer supplemented with protease inhibitor cocktail (Sigma Aldrich; Burlington, MA, USA). Total protein was quantified by bicinchoninic acid assay (BCA assay, Thermofisher Scientific; Waltham, MA, USA), normalized to 2 mg/mL and boiled with 5X SDS loading buffer. Lysates (30 μg) were resolved on a 4–20% Tris/glycine gel and transferred to a PVDF membrane. The membrane was probed with anti MYC (Cell Signaling; cat#5605; Danvers, MA, USA) and anti-GAPDH-HRP (Santa Cruz; sc-47724 HRP; Dallas TX, USA) antibodies.

ELISA: c-MYC degradation in L363 and H929 cells. L363 cells were grown in T-25 flasks in RPMI-1640 medium supplemented with 15% heat-inactivated fetal bovine serum (FBS) and 1% penicillin/streptomycin at 37 °C and 5% CO_2_. H929 cells were grown in T-25 flasks in RPMI-1640 medium supplemented with 10% FBS, 0.05 mM 2-mercaptoethanol, and 1% penicillin/streptomycin. Cells were treated with DMSO, TCH-165, or bortezomib (BTZ) at the indicated concentrations for 4 h (0.1% final DMSO concentration). Following the 4-hour treatment, cells were pelleted, washed with chilled PBS buffer, and resuspended in chilled lysis buffer (10 mM Tris, 100 mM NaCl, 1 mM EDTA, 1 mM EGTA, 1 mM NaF, 20 mM Na_4_P_2_O_7_, 2 mM Na_3_VO_4_, 1% Triton X-100, 10% glycerol, 0.1% SDS, and 0.5% deoxycholate; pH 7.4) supplemented with Complete Mini Protease Inhibitor Cocktail (Sigma Aldrich). The lysate was clarified by centrifugation at 5000 rpm (4 °C) for 15 min. Total protein of the lysate was quantified by bicinchoninic acid assay (BCA assay, Thermofisher Scientific; Waltham, MA, USA). and normalized to 1.5 mg/mL total protein. The supplied protocol for the human c-MYC (Total) ELISA Kit (Invitrogen; Waltham, MA, USA) was used to determine the total c-MYC concentration of the lysates. Prior to loading lysate into ELISA plate wells, lysate (5 µL) was diluted into 45 µL of supplied standard diluent buffer.

### 2.5. MYC-Luc Reporter Assay

The MYC Reporter (Luc)-HCT-116 Cell line, Catalog #60520, was bought from BPS Bioscience. MYC reporter (Luc)-HCT-116 cells were seeded in a white, clear bottom, 96-well microplate at 25,000 cells/well, in McCoy’s 5A media supplemented with 10% FBS and 1% penicillin/streptomycin. The plate was incubated at 37 °C with 5% CO_2_ overnight. The next day, cells were treated with 0.0 µM, 0.1 µM, 1.0 µM, 2.5 µM, 5.0 µM, 10 µM, or 20 µM TCH-165, in Opti-MEM supplemented with 0.5% FBS, 1% non-essential amino acids, 1 mM Sodium Pyruvate and 1% penicillin/streptomycin. The plate was incubated at 37 °C with 5% CO_2_ for 16 h. Firefly luminescence was measured using the One-Step Luciferase Assay System from BPS Biosciences.

### 2.6. Gene Expression Profiling of RPMI-8226 Cells

RPMI-8226 cells were treated with vehicle or TCH-165 (10 μM) for 4 h. Total RNA was obtained using the *mir*Vana miRNA Isolation Kit, with phenol (Thermofisher Scientific; AM1560; Waltham, MA, USA). Total RNA was quantified by Qubit (Thermofisher Scientific; Q32855; Waltham, MA, USA). and gene expression was evaluated against the nCounter^®^ PanCancer Pathways Panel^TM^ according to the standard NanoString protocol.

### 2.7. ROSALIND^®^ NanoString Gene Expression and Pathway Analysis

Data were analyzed by ROSALIND^®^ [36] with a hyperscale architecture developed by ROSALIND, Inc. (San Diego, CA, USA). Normalization, fold changes and *p*-values were calculated using criteria provided by NanoString. ROSALIND follows the nCounter^®^ Advanced Analysis protocol of dividing counts within a lane by the geometric mean of the normalizer probes from the same lane. Housekeeping probes to be used for normalization are selected based on the geNorm algorithm, as implemented in the NormqPCR R library [37]. Fold changes and *p*-values are calculated using the fast method as described in the nCounter Advanced Analysis 2.0 User Manual. The *p*-value adjustment is performed using the Benjamini–Hochberg method of estimating false discovery rates (FDR). Clustering of genes for the final heatmap of differentially expressed genes was done using the PAM (Partitioning Around Medoids) method using the fpc R library [38]. Hypergeometric distribution was used to analyze the enrichment of pathways, gene ontology, domain structure and other ontologies. The topGO R library [39] was used to determine local similarities and dependencies between GO terms in order to perform Elim pruning correction. Several database sources were referenced for enrichment analysis, including Interpro [40], NCBI [41], MSigDB [42,43], REACTOME [44], WikiPathways [45]. Enrichment was calculated relative to a set of background genes relevant to the experiment.

### 2.8. Assessment of Apoptosis

Apoptosis was evaluated with the Caspase-Glo 3/7 assay kit (Promega; Madison, WI, USA) according to the manufacturer’s instructions. Briefly, 10,000 cells were plated in each well of 96-well white 96-well plates and treated with either 0.1% DMSO (vehicle control), or TCH-165 (1, 4, 5, and 10 μM) for 24 h. Staurosporine (81590, Cayman), an apoptosis inducer, was used as a positive control at 10 μM concentration for 3 h of incubation. After the incubation, the caspase reagent was added to the cells and incubated at 37 °C for 30 min. The luminescent signal was detected by the EnVision plate reader (PerkinElmer; Waltham, MA, USA) and the data were analyzed by Graph Pad Prism (9.2.0, Graph Pad Software Inc., San Diego, CA, USA).

### 2.9. Pharmacokinetic Assessment of TCH-165

The PK study was carried out with CD-1 mice (males, 6–8 weeks of age, n = 3 per treatment group) weighing between 24.9 g to 33.7 g immediately prior to dosing. TCH-165 (100 mg/kg) was administered by oral gavage, as a homogeneous suspension in a 3:7 (*v*/*v*) propylene glycol: 5% dextrose as a vehicle. A second dose of 100 mg/kg TCH-165 was administered approximately 8 h post-initial gavage. Blood was collected at 0.5 h, 1 h, 2 h, 4 h, 8 h, 9 h, 10 h, 12 h, and 16 h post-initial gavage in lithium heparin tubes. Plasma was separated by centrifugation and analyzed using mass spectrometry.

### 2.10. Pharmacokinetic Assessment and Tolerance in Dogs

Adult beagle dogs (male, 12 months of age, n = 3), weighing approximately 10 kg, received a single oral dose of the test compound at 500 mg bid delivered in capsule form. Blood (approximately 2.5 mL) was collected into lithium heparin tubes at 0, 1 h, 2 h, 4 h, 8 h, and 24 h post-day 1 and post-day 5 dose for PK assessment using mass spectrometry. Blood (approximately 2 mL) was also collected and divided into EDTA and standard clot tubes on day 6 post-dose for determination of CBC and clinical chemistry parameters, respectively, and analyzed at the Michigan State University Veterinary Diagnostic Laboratory. Clinical observations were conducted daily on non-dosing days and twice on dosing days.

### 2.11. Proteasome Activity in Canine PBMC Lysate

Canine blood samples obtained in BD Vacutainer CPT mononuclear cell preparation tubes containing sodium citrate were used to isolate PBMCs. For PBMC isolation, sample tubes were inverted five times and equilibrated for the centrifuge with sterile PBS buffer. Tubes were centrifuged at 1800× *g* with the acceleration set at 5 and a break set to 0 at room temperature for 30 min. Half of the plasma layer was discarded. Using a sterile Pasteur pipette, the buffy coat was collected and put into a 15 mL conical tube and sterile PBS buffer was added to 10 mL. The conical tubes were centrifuged at 300× *g* for 10 min and the supernatant was removed. Cells were resuspended in 150 µL of lysis buffer (50 mM Tris-HCl, 2 mM Na_2_ATP, 5 mM MgCl_2_, 0.5 mM EDTA, 10% glycerol) and lysed by vortex. Samples were centrifuged for 20 min at 14,000× *g*. The total protein concentration of the supernatant was determined using bicinchoninic acid assay (BCA assay), and the samples were normalized to 1 mg/mL. Samples were diluted to 0.036 µg/µL in assay buffer (38 mM Tris, 100 mM NaCl, pH 7.8) and 140 µL of the diluted sample was added to three wells of a black, clear-bottom 96-well plate. Substrate stock solution (10 µL of 375 µM Suc-LLVY-AMC in assay buffer) was added to each well. Fluorescence was measured and kinetic readings were taken every 5 min at 37 °C, at 380/440 nm for 1 h.

### 2.12. RPMI-8226 Xenograft Tumor Model

CB17-SCID mice (C.B-Igh-1b/IcrTac-Prkdcscid) (female, 3–4 weeks of age) were injected subcutaneously with 1.2 × 10^7^ cells per animal. Animals were randomized prior to inoculation and regrouped prior to treatment to ensure equal average tumor volumes per group. Treatment was initiated when tumors reached ~50 mm^3^. The treatment group (size n = 5) is compared to the control group (size n = 4). No data points were excluded from the study. Tumor dimensions were measured three times weekly using digital calipers. Body weights were measured three times weekly. Outcome measures were defined by tumor volume. Tumors were measured twice weekly with digital calipers. Tumor volume was calculated using the equation: tumor volume (mm^3^) = 0.5 * (length * width^2^). TCH-165 (150 mg/kg on day 1, reduced to 100 mg/kg on day 3 and for the remainder of the study), was prepared as a homogeneous suspension in a 3:7 (*v*/*v*) propylene glycol: 5% dextrose vehicle. Bortezomib (0.375 mg/kg day 0, 0.18 mg/kg day 2, 0.09 mg/kg day 5 and beyond) was given 3X per week intravenously. Bortezomib was reduced to 0.09 mg/kg per treatment for the remainder of the study, due to weight loss at higher doses. Using the R package Stats, a Kruskal–Wallis test was applied to the data, resulting in a significant *p*-value (*p* = 0.0286), indicating a significant difference in tumor volume across the three treatments. This was followed by the application of a pairwise Wilcoxon rank-sum test with an adjusted *p*-value via the Benjamini–Hochberg method, significance set at FDR < 0.05. The TCH-165 treatment was found to be significantly different from both the control and bortezomib groups, with an FDR of 0.0014 and 0.0127, respectively.

## 3. Results

### 3.1. TCH-165 Enhances the Proteolytic Activity of the 20S Proteasome

The 20S proteasome is a barrel-shaped multi-subunit protease with 28 subunits arranged in four stacked heptameric rings (Figure 1A). Its inner β-rings contain three catalytic subunits (β5, β2 and β1) that exhibit chymotrypsin-like (CT-L), trypsin-like (Tryp-L) and caspase-like (Casp-L) activities [46]. Its outer α-rings guard entrance into the proteolytic core via an allosterically-controlled gate-opening/closing mechanism, which is required for substrate access to the proteolytic sites [13]. In a previous mechanistic study, TCH-165 increased substrate accessibility to the 20S catalytic chamber through the 20S gate opening (Figure 1A) [21]. TCH-165 and other 20S activators identified in our lab were reported to only enhance the degradation of endogenous 20S substrates [16,17,19,21,22]. Structured proteins normally targeted by the 26S-ubiquitin-dependent proteasome system were not affected at concentrations of <30 μM, after which TCH-165 outcompeted the docking of the 19S regulatory particles. To determine the effective concentration (EC) at which TCH-165 doubles (200%) 20S activity (referred to hereafter as EC_200_), and the maximum fold enhancement of 20S activity (referred to hereafter as Fold^M^), the hydrolysis of canonical proteasome peptide substrates (Figure 1B) was measured in the presence of TCH-165. The concentration at which the compound double proteasome-mediated proteolysis (EC_200_) is a slightly different number, but preferred, over its previously reported EC_50_ value [21], measured the compound concentrations at which half of the maximum fold-increase in 20S proteolytic activity was reached. The EC_200_ allows for a more accurate comparison between different compound classes that may exhibit different maximum fold activities. The EC_200_ for the hydrolysis of the CT-L substrate, Suc-LLVY-AMC was 1.5 μM, with a Fold^M^ of 810%. The EC_200_ and Fold^M^ for the Tryp-L and Casp-L sites were also determined to be 2.7 μΜ (Fold^M^ 500%) and 1.2 μM (Fold^M^ 1290%), respectively.

### 3.2. 20S Proteasome Activation by TCH-165 Regulates MYC Degradation

Considering the critical role of MYC in driving tumor growth and relapse in hematological cancers, the effects of TCH-165 on the protein levels of MYC were evaluated in RPMI-8226, multiple myeloma cells. For these assays, RPMI-8226 was treated with TCH-165, with or without the proteasome inhibitor, bortezomib (BTZ), for 4 h. Treatment of RPMI-8226 cells with TCH-165 (5 μM) reduced MYC protein levels within 4 h (Figure 2A) compared to the vehicle control. Importantly, the efficacy of TCH-165 was nearly completely abrogated by the blocking of the proteasomes’ catalytic activities using the covalent proteasome inhibitor BTZ (5 μM). These observations implicate the proteasome as the likely target for the effect of TCH-165 on MYC and eliminate the possibility that the observed results are due to a decrease in the 26S proteasome activity [21] or an effect on other proteases, such as calpain-dependent MYC degradation [4]. In addition, TCH-165-induced clearance of MYC in various other multiple myeloma cells, including L363 (Figure 2B) and H929 (Figure 2C).

### 3.3. 20S Proteasome Activation by TCH-165 Regulates MYC-Mediated Gene Transcription

The ability of TCH-165 to modulate MYC-mediated gene transcription was evaluated using a luciferase reporter assay in HCT-116 cells with a stably transfected MYC-luciferase gene (Figure 3). Treatment of the cells with various concentrations of TCH-165 resulted in a concentration-dependent decrease of MYC-mediated luciferase transcription at an EC_50_ of 2.57 μM (Figure 3A, 95% CI 2.46–2.95). The concentrations of TCH-165 needed to inhibit MYC-mediated luciferase transcription, correlated well to the concentrations needed to reduce MYC protein levels. Importantly, the addition of bortezomib abrogated the effects of TCH-165 mediated MYC transcription and restored luciferase transcription back to the levels of the vehicle control (Figure 3B). This data further implicates the proteasome as being responsible for the reduction in MYC signaling by TCH-165.

### 3.4. Gene Expression Profiling Indicates Limited Changes in Gene Expression following TCH-165 Treatment

The consequences of small molecule 20S activation in RPMI-8226 cells on the expression of genes in key cancer pathways were evaluated using the NanoString nCounter system with the PanCancer Pathway panel (Figure 4A) and analyzed using the ROSALIND platform. Results of three biological replicates for each condition are presented here for the RPMI-8226 cells. This multiplexed transcript detection assay analyzed the expression of 770 genes from 13 cancer-driving pathways. The gene profile of the vehicle control (DMSO) was compared to specific changes in gene expression induced by TCH-165 in RPMI-8226 cells. Gene expression was normalized using the geometric mean of the most stably expressed housekeeping genes in the panel in order to control for sample input variability. All of the normalized gene expression data from this NanoString panel are presented in Appendix A.

The 20S proteasome enhancer TCH-165 affected the expression of only a few genes. Of the 730 endogenous genes which were detected, 32 were found to be significantly upregulated and 16 were found to be significantly downregulated, using a 1.5-fold change and *p*-adjusted value of <0.05 as cut-off values, as indicated in the volcano plot in Figure 4A. Gene set enrichment analysis conducted with ROSALIND revealed a transcriptional misregulation pathway having the highest directed significance score at −9.6, as shown in the heatmap in Figure 4B. This was followed by JAK-STAT and Wnt pathways, at −3.59 and -3.6, respectively. DDIT3, the DNA damage inducible transcript 3, which codes for the protein CHOP, enhances gene transcription via interaction with JUN and FOS and induces apoptosis following endoplasmic reticulum stress, showed the highest upregulation [47]. Similarly, DDIT4, another DNA damage inducible transcript, which inhibits the mammalian target of rapamycin complex (mTROC1), was also overexpressed. Of note, the mRNA levels of the heat-shock protein, HSPA6, were also significantly upregulated, likely indicating unfolded protein stress.

### 3.5. Various Multiple Myeloma Cancer Cell Lines, and Patient Derived Primary and Refractory Multiple Myeloma Cells Are Vulnerable to the Proteasome Activator, TCH-165

To evaluate the efficacy of TCH-165, the viability of RPMI-8226, L363 and NCI-H929 multiple myeloma cells was evaluated following treatment with TCH-165. TCH-165 inhibited the proliferation of these cells with low single-digit micromolar potency with a 50% cytotoxic concentration (CC_50_) of 0.9 μM, 5.0 μM and 4.3 μM in RPMI-8226, L363 and NCI-H929cells, respectively, after 72 h (Table 1, Appendix A).

Although bortezomib and second-generation proteasome inhibitors remain first-line therapies in multiple myeloma treatment, the lack of response and relapse by a large population of patients remains a challenge [43,44]. Accordingly, we evaluated the effect of TCH-165 and bortezomib (BTZ) in primary cells isolated by CD138+ enrichment from a newly diagnosed patient and a BTZ refractory MM patient. As shown in Table 1, the CC_50_ value of TCH-165 in cells from a newly diagnosed MM patient was 1.0 μM (95% CI 0.60–1.51 μM). Even the primary cells from an MM patient refractory to BTZ were sensitive to TCH-165 at single-digit micromolar potency (CC_50_ 8.1 μM; 95% CI 7.08–9.03 μM, Appendix A). Consistent with this patient being refractory to BTZ, BTZ was ineffective against the isolated primary cells from the refractory MM patient (CC_50_ >1 μΜ). Additional samples from larger populations will need to be investigated to determine whether 20S proteasome enhancers are generally effective in refractory patients. Unfortunately, the mechanism of resistance in relapsed MM patients is complex and still ill-defined, thus the exact mechanism by which 20S proteasome activators may overcome resistance is still unclear.

### 3.6. Pharmacokinetic Properties and Anti-Tumor Efficacy of 20S Enhancer in In Vivo Xenograft

Following the cellular data, we initiated pharmacokinetic (PK) studies to determine plasma concentrations of TCH-165 following oral administration. CD-1 mice were exposed to TCH-165 (100 mg/kg) by oral gavage, as a homogeneous suspension in a 3:7 (*v*/*v*) propylene glycol: 5% dextrose as the vehicle. A second dose of 100 mg/kg TCH-165 was administered 8 h post-initial gavage. Blood was collected at various time points following post-initial gavage and the AUC was calculated using the trapezoidal method with mean values from data collected from 0–16 h post-initial dose. Following an oral administration of TCH-165 (100 mg/kg), the Cmax following the first dose was 932.3 ± 184.5 nM; with a Tmax at 2 h. The Cmax following the second dose was 1434.0 ± 625.2 nM with a Tmax of 12 h. The mean AUC (0–16 h) was 13,975 h*nM/mL (Figure 5A and Appendix A). As esters can be prone to hydrolysis by esterases, it is important to note that the ester moiety was (by LC-MS) not metabolically labile, likely due to its very sterically demanding location [30].

Considering that the concentration at which TCH-165 doubles the proteolytic activity of the 20S proteasome (Figure 1B, EC_200_ = 1.5 μM) is similar to the effective concentrations in the RPMI-8226 cell line (Table 1, CC_50_ ~1 μM), as well as the maximum plasma concentration reached in mice (Figure 5A), we evaluated the efficacy of this oral dose in an in vivo RPMI-8226 xenograft model. Figure 5B clearly shows that a dose of 100 mg/kg TCH-165, blocks tumor growth and is associated with a minimal loss of body weight (<10%), indicating good tolerance (Appendix A). After 42 days, the vehicle control group had an average tumor volume of 1253.4 (±371.86) mm^3^, bortezomib 771.41 (±129.35) mm^3^ and TCH-165 304.43 (±49.50) mm^3^. Compared to the vehicle control, TCH-165 treatment by oral gavage resulted in a statistically significant reduction of tumor growth (75.71%, *p* < 0.05) (Appendix A).

### 3.7. In Vivo Tolerance and Target Engagement of the 20S Proteasome Enhancer, TCH-165

Following these initial results of efficacy and tolerance, we examined TCH-165 for PK and tolerance in dogs. Adult male beagles received TCH-165 (oral capsule, 50 mg/kg, bid) for five consecutive days to match the AUC and Cmax as the (minimum) effective dose in mice (Figure 5A,B). This dose (50 mg/kg bid) resulted in an AUC (0–24 h) of hnM/mL with a Cmax of 1.6 μM and a Tmax at 24 h (Figure 5A and Appendix A). Clinical observations, body weight monitoring (<1% change), as well as the standard panel of complete blood counts (Appendix A) and clinical chemistry panel (Appendix A) following five days of treatment, identified no significant changes compared to the pre-dose status. Target engagement studies using adult male beagles confirmed the enhancement of proteasome activity following TCH-165 treatment. Peripheral blood mononuclear cells (PBMCs) were isolated from blood samples of untreated and treated (TCH-165, 1920 mg BID, n = 2 per group) canines. PBMCs were lysed and evaluated for hydrolysis of the CT-L substrate, Suc-LLVY-AMC and drug concentration using mass spectrometry, showing a 488% (1.4 μM) and 233% (1.1 μM) significant increase in substrate proteolysis (*p* < 0.001), in the treated canines compared to untreated canines (Appendix A). These studies indicate that the 20S proteasome enhancer, TCH-165, appears to be well tolerated in vivo, at concentrations where it significantly enhanced proteasome activity.

## 4. Discussion

Multiple myeloma treatment continues to be very challenging despite several key advances in the last decade. These advances include the FDA approval of the proteasome inhibitors bortezomib [48], carfilzomib [49] (and ixazomib for some patients) [50] as front-line therapies for MM, as well as the addition of anti-CD-38 antibody, daratumumab [51]. Most recently, a B-cell maturation antigen directed genetically modified autologous chimeric antigen receptor (CAR)T cell therapy (idecabtagene vicleucel) has been approved, with breakthrough therapy designation for multiple myeloma patients with relapsed or refractory MM after four or more lines of prior therapy [52]. While the 5-year survival of 30–50% as of 2016 [53,54] has improved for some patients that responded to the newer therapies, relapse is still very common [55].

MYC is a transcription factor that activates the expression of many pro-proliferative genes following its dimerization to the MYC-associated factor X (MAX). Overexpression of MYC is one of the main oncogenic drivers in most hematological malignancies, including MM. Importantly, MYC overexpression not only drives cancer growth, but it also induces oncogenic transformation, chemoresistance and disease relapse [56]. MYC is a highly unstable protein that is regulated by the ubiquitin-proteasome system (UPS), following a series of post-translational modifications [4,11]. However, recent studies have also implicated the ubiquitin-independent degradation of MYC through the 20S proteasome associated with its activator complex, REGγ (11S cap) [14]. Interestingly, overexpression of REGγ was found to enhance MYC protein degradation, and subsequently inhibit MYC-mediated gene expression and cell proliferation [14].

We previously reported the use of TCH-165 as a small molecule activator of the 20S proteasome [21]. Considering the parallels to 20S proteasome activation with REGγ, we evaluated the efficacy of our previously reported 20S proteasome enhancer, TCH-165, on the induction of MYC degradation. We found that MYC protein levels were rapidly reduced (within 4 h) in a concentration-dependent manner in the multiple myeloma cell lines, RMPI-8226, L363 and NIH-H929 cells, in response to 20S proteasome activation by TCH-165 (Figure 2). Importantly, blocking the catalytic activity of the proteasome with the proteasome inhibitor, bortezomib, abrogated all effects of TCH-165. Specifically, as shown in (Figure 3 and Figure 4), bortezomib prevented both TCH-165-induced degradation of MYC, as well as TCH-165 inhibition of MYC-mediated gene transcription, clearly implicating the proteasome as the likely protease responsible vs. other proteases, such as calpain-dependent MYC degradation.

The mechanism of 20S proteasome activation by TCH-165 involves the modulation of the assembly of the 26S proteasome, favoring the proteolytically active conformation of the 20S proteasome [21]. Moreover, 26S proteasome activity was not affected by TCH-165 [21]. We showed in various cell lines, including RPMI-8226, that only at high concentrations of TCH-165 (>30 μM), disassembly of the 26S proteasome inhibited 26S-mediated degradation of ubiquitinylated substrates (i.e., analogous to 26S proteasome inhibition) [21]. To determine whether the reduction of MYC proteins levels by TCH-165 was due to a decrease in 26S proteasome-mediated proteolysis or due to an increase in activity of other proteases, cells were treated with the proteasome catalytic site inhibitor, bortezomib (BTZ). The results shown in Figure 2 and Figure 3 illustrate that the effects of TCH-165 on MYC protein levels as well MYC-mediated gene transcription are abrogated by BTZ, thereby indicating that MYC clearance induced by TCH-165, is likely a 20S proteasome-mediated event.

In addition to the direct degradation of MYC by the enhancement of 20S proteasome activity, the possibility of alternative pathways of MYC clearance, following TCH-165 treatment, remains possible. For example, MYC proteolytic stability and MYC transcriptional activity are highly regulated by multiple E3 ligases [4], many of which are intrinsically disordered, overexpressed in cancer, and positively regulate MYC transcription [57]. Enhancing 20S proteasome activity may also indirectly impact MYC clearance by affecting its many post-translations modifications that regulate its proteolytic stability or via its interaction with its disordered binding partner MAX [58,59].

Among the genes included in the PanCancer Pathway, those relating to gene misregulation were the most upregulated, especially DDIT4. Relatively few genes showed differential expression upon TCH-165 treatment. It is possible that the relatively short exposure used in the gene expression study was not sufficient to elicit this effect Even though relatively few genes were affected by TCH-165 treatment, the cells differentially expressed multiple MYC target genes, including TCF3, DDIT3, DDIT4 and VEGFA, and resulted in induction of apoptosis. This is also consistent with the cytotoxic effects observed with TCH-165 treatment in Table 1 and Appendix A.

Overcoming resistance is critical in the search for new therapeutic approaches. The 20S proteasome enhancer, TCH-165, was found to be effective in primary cells from a patient who was refractory to bortezomib treatment. Whether or not the decrease in MYC protein by 20S proteasome enhancement is responsible for overcoming this resistance is not currently known, but it is an intriguing possibility that warrants further investigation.

The translational efficacy of the 20S proteasome enhancer was evaluated in an RPMI-8226, multiple myeloma xenograft model. TCH-165, when administered to mice at 100 mg/kg twice daily, resulted in plasma concentrations approximating the AC_200_. At this dose, significant inhibition of tumor growth was observed (Figure 5A,B) in the mice. The treatment was well tolerated by the mice, with <10% loss of body weight, and resulted in a 76% reduction of tumor growth. More detailed tolerance of 20S enhancement by TCH-165 was examined in beagles (n = 3) by approximating the AUC and Cmax of the mouse study, with TCH-165 treatment for five consecutive days (oral capsule, 50 mg/kg, bid). Clinical observation, body weight (<1% change), as well as the standard complete blood count (37 criteria) and clinical chemistry (29 criteria) following this five-day tolerance study, identified no significant changes compared to the pre-treatment evaluations (Appendix A). The only notable change in the clinical chemistry profile observed in the tolerance study in dogs was an increase in plasma iron concentrations on day 5 of TCH-165 administration. A review of proteins involved in iron transport and storage did not reveal any intrinsically disordered proteins. Plasma iron levels are mediated by hepcidin, which, in broad terms, acts as a negative regulator of iron in circulation by increasing endocytosis of ferroportin, increasing iron in cells and decreasing iron in the plasma [60,61]. Hepcidin expression is induced by iron overload, infection and inflammatory cytokines. Interestingly, the transcriptional activation of hepcidin was found to depend on USF1/ USF2 and MYC/MAX heterodimers through E-boxes within its promoter [62]. Therefore, one can speculate that decreases in MYC expression may lead to decreases in hepcidin levels, resulting in increased levels of iron in the plasma. Plasma iron is mostly bound to transferrin, which is usually only 30% saturated [63]. Thus, the increase in plasma iron observed may be well tolerated. In addition, MYC is a negative regulator of expression of H-ferritin and a positive regulator of iron-responsive element-binding protein 2 and thus may have complex effects on iron handling [64]. This observation requires further mechanistic studies in relevant tissues in future studies.

These studies demonstrated the first translational anti-tumor efficacy and acceptable in vivo tolerance of the 20S activator, TCH-165, and support further pre-clinical investigations in MM disease models that more accurately recapitulate hematological diseases [65]. Overall, the active concentration at which TCH-165 doubles 20S proteasome activity (EC_200_ 1.5 μM) corresponded well to the effective concentration required to reduce MYC gene transcription (EC_50_) and induce 50% cell death (CC_50_) in various cell cultures, and corresponded to the Cmax obtained in vivo and required to inhibit tumor growth.

## 5. Conclusions

Our data indicate that TCH-165 enhances the activity of the 20S proteasome and reduces MYC protein levels, which may be responsible for its anticancer efficacy. However, the role of other disordered protein targets cannot be excluded, nor can the enhanced degradation of MYC be assumed to be solely responsible for its anticancer efficacy. While further mechanistic studies are needed, TCH-165 and enhancement of the 20S proteasome activity clearly pave a new avenue for anti-tumor activity in MM. Importantly, the 20S proteasome enhancer was well tolerated both in mice and in canines, validating this new approach for further therapeutic assessment for various MYC-driven cancers.

## 6. Patents

U.S. Serial No., 17/490,999 “Treatment of Malignancies”; Vilma Yuzbasiyan-Gurkan and Jetze J. Tepe, inventors; Board of Trustees Michigan State University, assignee. The patent application is currently pending. 

## Figures and Tables

**Figure 1 biomedicines-10-00938-f001:**
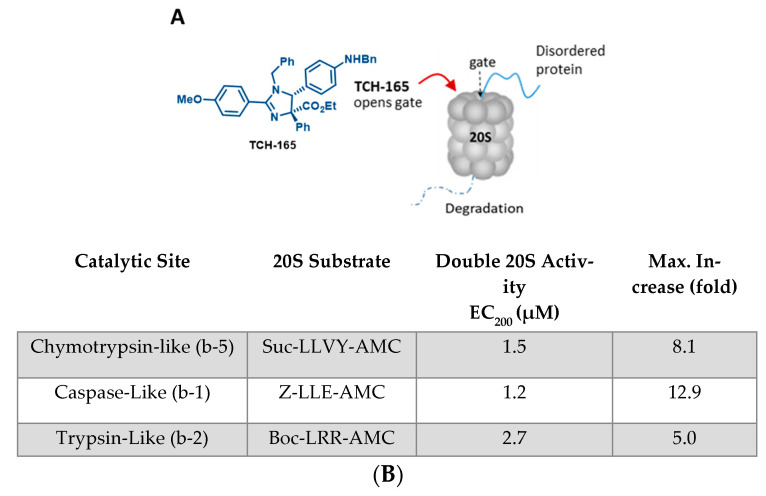
(**A**) Structure of TCH-165 and its proposed model of α-ring binding, allowing the gate to open for easy access of disordered proteins to the internal proteolytic sites; (**B**) EC_200_ values of TCH-165 and maximum fold enhancement of 20S activities for each of the three catalytic sites.

**Figure 2 biomedicines-10-00938-f002:**
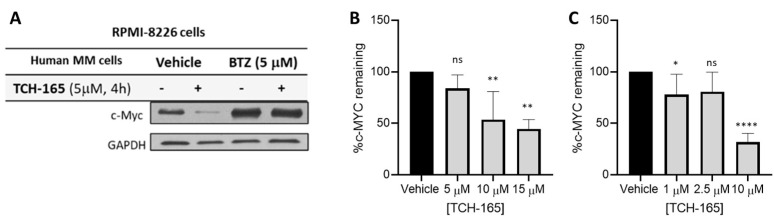
(**A**) Multiple Myeloma (RPMI-8226) cells treated with vehicle or TCH-165 (4 h, 5 mM) in the presence and absence of proteasome inhibitor BTZ (5 μM); (**B**) percent MYC remaining in L363 cells treated with various concentrations of TCH-165 (n = 3); (**C**) percent MYC remaining in H929 cells treated with various concentrations of TCH-165 (n = 4). * *p* < 0.05, ** *p* < 0.01, **** *p* < 0.0001).

**Figure 3 biomedicines-10-00938-f003:**
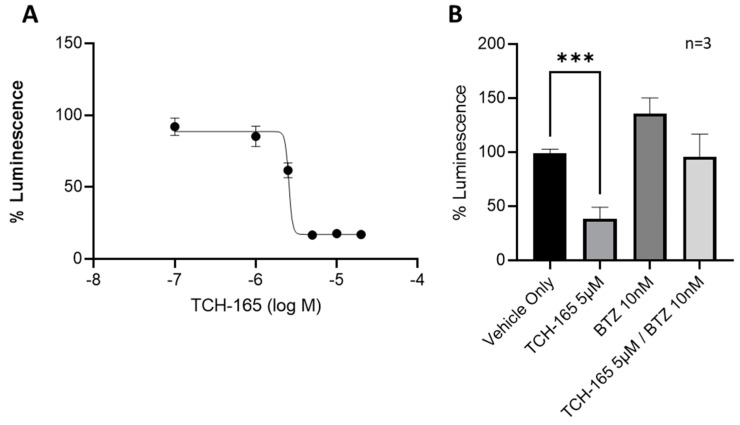
MYC-luciferase reporter assay: (**A**) MYC-luciferase reporter assay in stable transfected HCT-116 cells treated with various concentrations of TCH-165 (EC_50_ 2.57 μM; 95% CI 2.46–2.95); (**B**) MYC-luciferase reporter assay in stably transfected HCT-116 cells treated with vehicle, TCH-165 (3 μM), bortezomib (BTZ, 10 nM) and bortezomib (2 h pre-incubation, 10 nM) followed by TCH-165 (3 μM). *** *p* < 0.001.

**Figure 4 biomedicines-10-00938-f004:**
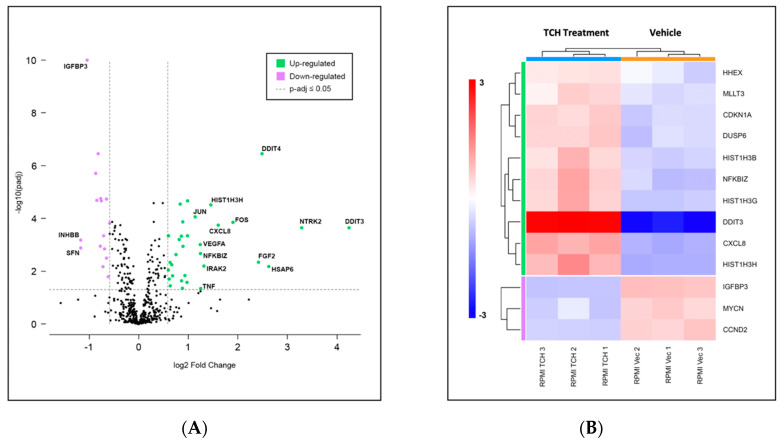
(**A**) Volcano scatter plots of genes in cancer pathways analyzed by the PanCancer multiplexed transcript detection assay. Dotted vertical lines indicate a 1.5-fold change in normalized gene expression, with overexpressed genes in green and under-expressed genes in purple. Labels indicate genes showing over 2-fold change; (**B**) Changes in gene expression observed over three independent experiments indicating transcriptional misregulation are presented in the heatmap. Consistent with the upregulation of the DNA damage response, we demonstrated induction of apoptosis in all three MM cell lines (Appendix A).

**Figure 5 biomedicines-10-00938-f005:**
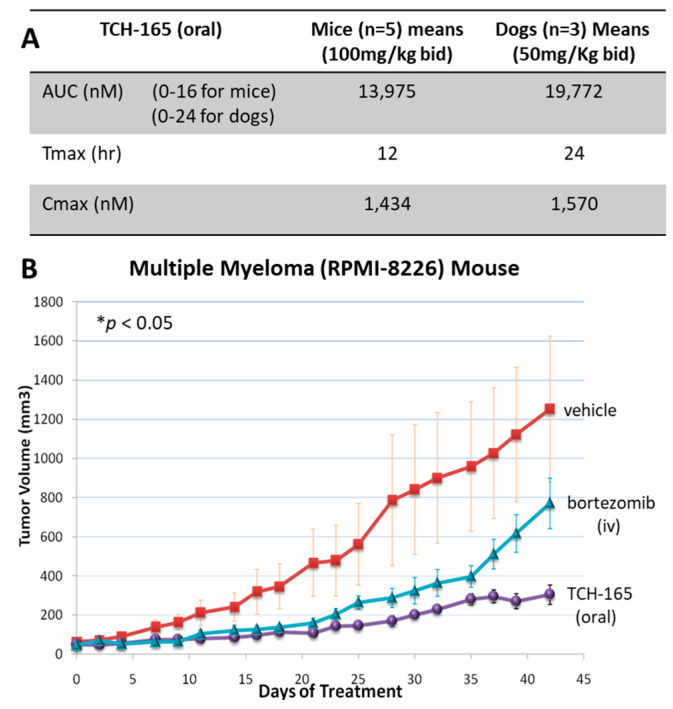
In vivo efficacy and tolerance of the proteasome activator, TCH-165. (**A**) Comparative analysis AUC, Tmax and Cmax of TCH-165 bid in mice (n = 5) and dogs (n = 3), calculated using PKSolver [45]. (**B**) RPMI-8226 subcutaneous xenograft model in SCID mice treated with vehicle (3:7 (*v*/*v*) propylene glycol/5% dextrose), and TCH-165 (100 mg/kg, bid with oral gavage), bortezomib (at 0.375 mg/kg day 0, 0.18 mg/kg day 2, 0.09 mg/kg day 5 and beyond) was given 3X per week intravenously. Tumor volumes show a significant difference between TCH-165 vs. control as well as TCH-165 vs. bortezomib groups at FDR of 0.0014 and 0.0127, respectively, using the pairwise Wilcoxon rank-sum test with the Benjamini–Hochberg adjustment, after application of Kruskal–Wallis test showing significant *p*-value of 0.0286 across all treatments, calculated using R Stats v3.6.2.

**Table 1 biomedicines-10-00938-t001:** CC_50_ values of MM cell lines and patient cells. Cytotoxicity of TCH-165 in human RPMI-8226 (CC_50_ 0.9 μM; 95% CI 0.8–1.2μM), L363 (CC_50_ 5.0 μM; 95% CI 4.1 −5.1 μM), NIH-H929 (CC_50_ 4.3 μM; 95% CI 2.2.8–6.6 μM) cells after 72 h treatment, as well as primary human MM cells of newly diagnosed patient (CC_50_ 1.0 μM; 95% CI 0.6–1.5), relapsed patient (CC_50_ 8.1 μM; 95% CI 7.1–9.0 μM). For comparison, primary human MM cells of newly diagnosed patient treated with BTZ (CC_50_ 4.0 nM; 95% CI 2.3–7.1), and BTZ was ineffective in a relapsed patient sample. CI was determined using a 4-parameter curve. * CI determined using a 3-parameter curve.

**MM Cells Treated with TCH-165**	**CC_50_ (μM) & 95% Confidence Interval (CI)**
RPMI-8226	0.9 (95% CI 0.8–1.2 μM)
L363	5.0 (95% CI 4.1 -5.1 μM)
NCI-H929	4.3 (95% CI 2.2–6.6 μM *)
**Cells Treated with TCH-165**	**CC_50_ (μM)**
Primary MM patient cells	1.0 (95% CI 0.6–1.5 μM)
Refractory MM patient cells	8.1 (95% CI 7.1–9.0 μM)
**Cells Treated with BTZ**	**CC_50_ (nM)**
Primary MM patient cells	4.0 (95% CI 2.3–7.1 nM)
Refractory MM patient cells	>1000 (95% CI N/A)

## Data Availability

The body of the manuscript and the Appendix A contain all of the data.

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
