# Peer review of "Small Molecule 20S Proteasome Enhancer Regulates MYC Protein Stability and Exhibits Antitumor Activity in Multiple Myeloma"

_biomedicines, 2022, doi:10.3390/biomedicines10050938_

Round 1

Reviewer 1 Report

Interesting paper from the Tepe group using TCH-165 to degrade Myc which has a significant effect on xenograft mouse models of MM. Overall the paper is logical and the data seems scientifically sound. I have just a few comments that need to be addressed.

-For the mice, did they use both sexes? Also for tumor volume measurements, how was this accomplished as MM don't typically cause solid tumors? Did they also not do survival plots? It is odd that Btz did a poor job of suppressing tumor volume as compared to TCH-165, did that translate to survival?

-For the EC200 experiments is it really relevant to compare the degradation rates of a 4-mer peptide when really the interest is in degrading Myc. Could the experiment of dosing cell lysate with different amounts of TCH-165 and blotting for myc not be performed to find a relevant EC200 concentration?

-The y-axis of Figure 3A has a spelling error.

-Regardless of Myc, why are there no viability curves? If TCH-165 is degrading Myc, which the cells are dependent on, can TCH-165 alone cause cell death? This would help with the impact of the paper.

Overall I am supportive of this publication after these issues are addressed.

Reviewer 2 Report

In this study, Evert Njomen and colleagues show that a small molecule (TCH-165) previously shown to “activate” 20S proteasome, also exhibits antitumor activity. They show that TCH-165 treatment significantly decreases c-myc levels through proteasome degradation. TCH-165 is shown to decrease the viability of MM patient cells, even in refractory cells that don’t respond to bortezomib and, in an in vivo xenograft model, TCH-165 was shown to reduce tumor growth. Importantly, TCH-165 is well tolerated when administered to mice and dogs.   

The conclusions of this study are well supported by the data, even if the antitumor activity of TCH-165 could not be directly linked to the decrease in MYC.

Overall, the manuscript is clear and well written. Data is adequately presented and the description of experimental procedures is sufficiently detailed.

There are only minor issues I would like the authors to address.

Minor points:

1- Am I correct in thinking that the metrics shown in fig.1B were obtained from proteolytic activity assays that were previously published (https://doi.org/10.1021/acs.biochem.8b00579). The Max Increase folds are exactly the same, only now a new metric is taken from that data ((EC200). I have no problem with this but I would prefer if this was made clear in the manuscript. 

2- Bortezomib was also shown to down-regulate c-myc (PMID: 26317903, PMID: 32724371), however, here it is used as a specific inhibitor of proteasome to rescue myc from degradation. Can the authors comment on this discrepancy? In this case, I think involvement of proteasome would be better shown with MG132. 

3- As per the authors, TCH-165 not only enhances 20S proteasome directly, but can also somehow alter the equilibrium between 20S and 26S particles favoring disassembly of the 19S regulatory particle. So its effect on 20S activity is two-fold. Do the authors have data suggesting this disassembly already occurs at the determined Cmax of 1.4 - 1.6 μM?

4- Can the authors extend their observations on why it is important to note that the ester moiety of TCH-165 is not metabolically labile? Does this point to a more prolonged effect of the drug? What are the implications? 

5- Can the authors speculate on the cause for the elevated iron levels? Since hemolysis is apparently normal, could this point to a specific iron-binding intrinsically disordered protein that is now overly degraded? Or an IDP involved in iron regulation?

6- Please give details on how or where the myc-luciferase cells were obtained.

Round 2

Reviewer 1 Report

All of my comments were addressed, looking forward to seeing this in print!